# Aquaporins in Renal Diseases

**DOI:** 10.3390/ijms20020366

**Published:** 2019-01-16

**Authors:** Jinzhao He, Baoxue Yang

**Affiliations:** State Key Laboratory of Natural and Biomimetic Drugs, Department of Pharmacology, School of Basic Medical Sciences, Peking University, Beijing 100038, China; 18200288030@163.com

**Keywords:** aquaporin, vasopressin, nephrogenic diabetes insipidus, acute kidney injury, diabetic nephropathy, polycystic kidney disease, renal cell carcinoma

## Abstract

Aquaporins (AQPs) are a family of highly selective transmembrane channels that mainly transport water across the cell and some facilitate low-molecular-weight solutes. Eight AQPs, including AQP1, AQP2, AQP3, AQP4, AQP5, AQP6, AQP7, and AQP11, are expressed in different segments and various cells in the kidney to maintain normal urine concentration function. AQP2 is critical in regulating urine concentrating ability. The expression and function of AQP2 are regulated by a series of transcriptional factors and post-transcriptional phosphorylation, ubiquitination, and glycosylation. Mutation or functional deficiency of AQP2 leads to severe nephrogenic diabetes insipidus. Studies with animal models show AQPs are related to acute kidney injury and various chronic kidney diseases, such as diabetic nephropathy, polycystic kidney disease, and renal cell carcinoma. Experimental data suggest ideal prospects for AQPs as biomarkers and therapeutic targets in clinic. This review article mainly focuses on recent advances in studying AQPs in renal diseases.

## 1. Introduction

Aquaporins (AQPs) are a family of highly selective transmembrane channels that mainly transport water across the cell and some facilitate low-molecular-weight solutes. AQPs consist of 13 members (AQP0–AQP12) in mammals and are widely distributed in various tissues and organs. According to their primary structure, they have been classified into three subfamilies. Water-selective AQPs include AQP0, AQP1, AQP2, AQP4, AQP5, AQP6, and AQP8, which are also known as orthodox aquaporins. Aquaglyceroporins, including AQP3, AQP7, AQP9, and AQP10, are permeable to water and some small uncharged solutes, such as glycerol and urea [1,2]. AQP3 AQP8, and AQP9 have been demonstrated to transport hydrogen peroxide (H_2_O_2_) in mammalian cells [3,4]. The third subfamily named superaquaporins, including AQP11 and AQP12 [5], has low homology at its amino acid level with other classical AQPs. Due to a wide spectrum of pathophysiological function in balancing water homeostasis, modulating intracellular signaling, and regulating cell proliferation and oxidative stress response, AQPs have been proven to participate in renal diseases, dermatosis, bowel disease, and cancer [3,6,7,8]. This review article mainly focuses on the effect of AQPs in kidney diseases.

## 2. Expression of Physiological Function of AQPs in the Kidney

In the kidney, eight AQPs, including AQP1, AQP2, AQP3, AQP4, AQP5, AQP6, AQP7, and AQP11, are expressed in different segments and various cells to maintain normal urine concentration function, tissue development and substance metabolism [9,10] (Figure 1).

AQP1 is the first discovered water channel, and is located in the apical and basolateral plasma membrane of the proximal tubule, descending thin limbs of Henle, and descending vasa recta to mediate water reabsorption [11]. AQP1 is a highly selective water-permeable channel. Mice defective with AQP1 exhibit polyuria, indicating its key role in the formation of hypertonicity [12,13].

AQP2 is one of the most important channel proteins involved in regulating urine concentration, and is located at the apical membrane of principal cells in the collecting duct [14]. The water reabsorption function of AQP2 is mainly regulated by arginine vasopressin (AVP) via increasing intracellular production of cyclic adenosine monophosphatase (cAMP) and further phosphorylation of AQP2 at Ser256 and Ser269 to stimulate the intracellular trafficking of AQP2 to the plasma membrane [15,16,17] (Figure 2). Besides AVP, researchers have demonstrated that activation of bile acid receptor TGR5 and hydrogen sulfide (H_2_S) stimulates the expression of AQP2 via cAMP-protein kinase A (PKA) signaling pathway and attenuates the defection of urinary concentration in mice [18,19]. In fact, AQP2 expression, phosphorylation, and trafficking are not only activated by canonical AVP/cAMP/PKA signaling but also other signaling. Erlotinib, an epidermal growth factor receptor (EGFR) inhibitor, has been reported to enhance AQP2 accumulation in plasma membrane and water reabsorption through increasing phosphorylation of AQP2 at Ser-256 and Ser-269 and reducing endocytosis of AQP2 without affecting classic PKA signaling [20]. Administration of Wnt5a alleviated decreased urine osmolality and upregulation of AQP2 via activation of calcium/calmodulin/calcineurin signaling [21]. Additionally, the expression of AQP2 can be regulated by several transcription factors, such as AP-1, NF-κB, and NFAT [22]. cAMP-responsive element binding protein (CREB) has been recognized as a major regulator of AQP2 expression. However, recent evidence identified that C/EBPβ is pertinent to transcriptional regulation of *AQP2* and the relationship between CREB and *AQP2* is indirect [23]. Notably, normal expression of AQP2 in apical plasma membrane plays a critically determinant role in renal urine concentration and body water balance. Deletion or mutation of the AQP2 gene causes severe water disorders and triggers the initiation of nephrogenic diabetes insipidus (NDI). Urinary excretion of AQP2 has been recognized as a useful marker for diagnosis of renal diseases [24]. 

AQP3 is constitutively located in the basolateral membrane of principle cells in the cortex and outer medullar collecting duct and is regulated by thirst, AVP, and aldosterone. AQP4 is mostly distributed in the basolateral membrane of principle cells of the medullary segment of the collecting duct. Protein kinase C and dopamine rather than AVP affect phosphorylation of AQP4 to regulate water permeability. AQP3 and AQP4 could export water entering cytoplasm via AQP2. Of note, AQP3 also facilitates glycerol and hydrogen peroxide transport through the cell membrane, which regulates a series of intracellular signaling and affects cellular functions, such as cell proliferation, apoptosis and migration [6]. AQP3-null mice showed NDI-like phenotype, while the absence of AQP4 only presented mild urinary concentration defect. However, the double knockout mice of AQP3/AQP4 have a greater impairment of urinary function than AQP3-null mice [25,26], which may be due to their similar localization and water permeability in the urinary tract.

A few years ago, scientists firstly reported that AQP5 is located in type B intercalated cells of collecting ducts [27]. AQP6 is localized in the intracellular vesicles of intercalated cells and colocalized with H^+^-ATPase [28]. AQP6 hardly transports water from the membrane unless at a low pH value. The function of AQP5 and AQP6 in the kidney is still not clear.

AQP7 is expressed in the brush border of the S3 segment of the proximal tubule, and shows great effect on metabolism by regulating the transportation of glycerol. Defective AQP7 expression has little effect on water permeability of proximal tubules, but is associated with significant metabolism disorders, like obesity and insulin resistance [29].

AQP11 is uniquely localized in the membrane of endoplasmic reticulum (ER) of proximal tubular cells. The transport function of AQP11 is controversial regarding whether it transports water and glycerol or only glycerol [30]. AQP11 knockout mice develop uremia due to the renal cysts derived from the proximal tubule. 

Nowadays, the pathophysiologic functions of AQPs in renal-specific cell types and liquid homeostasis have been deeply studied to provide the therapeutic targets. The data showed that AQPs might be an ideal biomarker for renal diseases [31,32,33]. This review article focuses on the pathophysiological effect of AQPs in renal diseases and potential therapeutic targets of AQPs.

## 3. Functional Deficiency of AQP2 Causes NDI

NDI is a rare disease characterized by polyuria and polydipsia. Patients with NDI produce around 12 L of urine per 24 h. Congenital NDI is caused by the mutation of the AVP receptor 2 (about 90%) or AQP2 (about 10%) gene [34]. AQP2 mutations in NDI patients are autosomal recessive inheritance. In patients, the urinary AQP2 excretion could be used to evaluate the mutation of AVPR2 [35]. Acquired NDI results from chemical substances, electrolyte abnormalities, and obstructive uropathy [14,36]. Lithium-induced NDI is the most common cause among patients, due to the wide use of lithium for psychiatric disorders, which affects AQP2 function. So far, the treatment for NDI in clinic mainly includes dietary modulation, thiazide diuretics, amiloride, and prostaglandin synthesis inhibitors [37]. However, these therapies only ameliorate the symptoms of NDI without radical cure. The alteration of AQP2 expression is the predominant cause of heritable and acquired NDI. Thus, strategies to increase the functional accumulation of AQP2 in the apical membrane might be useful in treating NDI, such as AVP analogues, prostaglandin receptor agonists, and cGMP phosphodiesterase inhibitors [34].

AQP2 functional deficiency causing NDI was determined by several genetic mouse models [38,39,40]. The AQP2-T126M mutant protein of AQP2^T126M/−^ mice kidney collecting duct epithelial cells was distributed in the endoplasmic reticulum rather than in the apical membrane where the wild-type AQP2 protein was located (Figure 3A). Meanwhile, less than 10% of collecting duct epithelial cells presented normal localization of AQP2 (Figure 3A, white arrow). Immunoblot analysis of wild-type and AQP2^T126M/−^ mice whole-kidney homogenates showed that the core-glycosylated form of AQP2-T126M, a band at ~31 kDa rather than wild-type AQP2, two bands around 34–40 kDa and 29 kDa, was shown in AQP2^T126M/−^ mice (Figure 3B). After treatment of heat shock protein 90 (Hsp90) inhibitor 17-allylamino-17-demethoxygeldanamycin (17-AAG), the immunoblot analysis of AQP2 showed the restoration of glycosylated and nonglycosylated bands and the lower level of ~31 kDa and ~23 kDa bands, which implied the partial correction of AQP2 expression in AQP2^T126M/−^ mice. The results showed that 17-AAG partially restored defective AQP2 cellular processing and increased urine osmolality in AQP2 mutation mice by >300 mosmol but had no effect in AQP2-null mice. (Figure 3D). This study not only generated an adult NDI animal model that mimics NDI patients with AQP2 mutation, but also used 17-AAG for partially correcting the expression of AQP2 to attenuate the urinary concentration defect. These data suggest a novel therapeutic way to treat NDI and show a promising clinic prospect for 17-AAG and other small molecules. Of note, regulating AQP2 protein expression has been an effective way to elucidate the underlying mechanisms of NDI and explore the potential drugs [41]. Recently, another study generated *Keap1*^−/−^::*Nrf2*^Flox/Flox^::*K5*-Cre mice in which constitutive Nrf2 activation caused NDI by reducing AQP2 expression [42].

In hypokalemia-induced NDI, autophagic degradation led to the decreasing expression of AQP2 in the early stage [43]. Based on proteomic and bioinformatic analysis, researchers revealed that the degradation of AQP2 plays a pivotal role in the onset of hypercalcemic-induced NDI [44]. Lithium-induced NDI has been a common animal model for exploring NDI-related mechanisms and potential drugs. Sustained intake of lithium caused the downregulation of AQP2 and AQP3 protein levels and impaired urinary concentration ability. Utilizing this model, scientists have provided evidences that activation of TGR5 and production of H_2_S attenuated the symptoms of NDI [18,19]. In the other study, researchers found that pharmacological blockade or genetic deletion of P2Y12 blunted the reduction of AQP2 and polyuria induced by lithium. Therefore, these data indicate that pharmacological activation of TGR5, H_2_S, and inhibition of P2Y12 show potential to treat NDI. 

In the past few years, PKA signaling has been identified as a traditional pathway for exploring underlying mechanisms and therapeutic targets of NDI. Deletion or inhibition of adenylate cyclase 6 or glycogen synthase kinase 3β (GSK3β) decreased the level of cAMP and reduced expression of AQP2 to elicit NDI [45,46]. Fumiaki et al. have reported that low-molecular-weight compound 3,3′-Diamino-4,4′-dihydroxydiphenylmethane (FMP-API-1) and its derivatives stimulated PKA and obviously increased AQP2 activity to protect mice from NDI [47]. Besides, prostaglandin E2 receptor 4 and (pro)renin receptor play important roles in regulation of AQP2 expression and urine concentrating capability via activating the cAMP/PKA pathway [48,49]. However, strategies targeting the cAMP/PKA pathway have shown severe side effects and poor outcomes. Alternative novel therapies have been sought in recent years, such as EGFR inhibition and Wnt5a administration. Statins, as a kind of safe and effective lipid-lowering drug, show potential to remedy NDI through increasing the mRNA and proteins of AQP2 [50]. 

Furthermore, extracellular matrix (ECM) affects the expression of AQP2 and might open new venues for treatment of NDI. Integrin-linked kinase (ILK) as a scaffold protein that links ECM with intracellular signaling increased the expression AQP2 by activating ILK/GSK3β/NFAT axis and knockdown of ILK displayed a symptom of NDI [51].

## 4. AQPs in Acute Kidney Injury

Acute kidney injury (AKI) is a worldwide syndrome with high morbidity and mortality characterized by rapidly declined glomerular filtration. The occurrence of AKI is approximately 13.3 million people per year and 36% of them are required to take renal replacement therapy. AKI can be elicited by prerenal factors (almost 60%; heart failure, sepsis, drugs), intrarenal factors (approximately 35%; nephrotoxic substances induced acute tubule necrosis, interstitial nephritis, intrarenal deposition), and postrenal factors (almost 5%; tumor, clot, neurogenic bladder) [52].

Renal ischemia/reperfusion (I/R) injury is a common cause of AKI. Several reports have shown that the expression of AQPs is closely related with I/R-induced AKI [53]. The protein expression of AQP1, AQP2, and AQP3 has been shown significantly decreased which could account for the defect in urinary concentration in I/R mice [54]. Urinary exosomal release of AQP1 and AQP2 was reduced at the stage of AKI and in bilateral and unilateral I/R rats [53]. Lack of AQP1 makes mice more sensitive to I/R kidney injury. A study reported that endotoxemia-induced AKI is more severe in AQP1 knockout mice, implying the importance of AQP1 channel [55]. 

Lei et al. proved that deletion of *AQP3* aggravated the kidney injury by increased apoptosis and inhibited MAPK signaling in I/R mice [56]. The results showed the blood urea nitrogen (BUN) and creatinine levels were significantly elevated after I/R in AQP3^−/−^ mice (Figure 4A,B). The SOD activity and MDA level were obviously altered in AQP3^−/−^ mice but not in wild-type mice after I/R (Figure 4C,D), which was in accordance with injured renal morphology and increased apoptosis level. Hematoxylin-eosin staining showed significant tubular morphological damage in AQP3^−/−^ kidneys with I/R, but not in wild-type kidneys (Figure 4E). For underlying mechanisms, Western blot analysis showed the activation of pro-apoptotic protein Bax and inhibition of antiapoptotic protein Bcl-2 and upregulated expression level of cleaved caspase-3 and p-p53, indicating the increased level of apoptosis in AQP3^−/−^ mice with I/R (Figure 4F, left panel). AQP3^−/−^ kidneys showed downregulated p38/ERK/JNK MAPK signaling, which was reversed by I/R. These data provide the evidence that AQP3-mediated apoptosis and MAPK signaling play pivotal roles in I/R, which might be an effective target in future. Global analysis of differential gene expression in desert-adapted animals showed that the expression of AQP4 is significantly reduced in acute dehydration, which might play a role in water handling to prevent kidney injury [57]. In the cisplatin-induced kidney injury animal model, the level of AQP1-3 decreased and L-carnitine ameliorated the urinary concentration defect by increasing AQP2 expression [58]. 

Another study has reported that downregulation of AQP2 expression in lipopolysaccharide-induced AKI participated in urinary concentration defect in sepsis [59]. Interestingly, reducing the activation of NF-κB signaling ameliorated the downregulation of AQP2 protein and sepsis-induced acute renal failure [60,61]. Using a renal transplantation animal model, the data showed that the reduced AQP2 expression in phase of acute graft rejection could be blunted by administration of cyclosporine [62,63]. Thus, the decreased level of AQP2 partially accounts for the mechanism of acute renal disease, which suggests targeting AQP2 as a new therapeutic way for AKI. In addition, AQP11 rs2276415 variant renders diabetic patients with higher risk for AKI [64,65]. To some extent, restoration or correction of AQPs may protect patients from AKI.

## 5. AQPs in Diabetic Nephropathy

Diabetic nephropathy (DN) is one of the most common complications of diabetes mellitus (DM), which occurs in one-third of diabetic patients [66]. DN is a slowly progressive disease with impaired renal function and fluid balance disorder, and is the leading cause of end-stage renal disease (ESRD) in adults [67]. As polyuria is an early sign of DN, the role of AQPs in the pathophysiology of DN has been studied in recent years [33].

A growing body of evidence showed that the expression of AQPs was dysregulated in DN patients. AQP5 is obviously increased in the kidney tissue of DN patients and urine AQP2 and AQP5 have been identified as potential novel biomarkers of diabetic nephropathy with sensitivity, early appearance, and low invasiveness features [68,69,70]. A community-based cohort study involving 620 participants with chronic kidney disease (CKD) firstly revealed that AQP11 rs2276415 variant is associated with CKD progression [69]. Similarly, Choma et al. reported AQP11 rs2276415 variant as a genetic factor predisposing type 2 diabetic patients to a greater risk for the development of CKD [64]. Early detection of urine exosomes and genetic screening of AQPs might assist clinical diagnosis and treatment of DN. 

## 6. AQPs in Polycystic Kidney Disease

Autosomal dominant polycystic kidney disease (ADPKD) is a common hereditary disease, affecting 1/1000–1/400 individuals worldwide [71]. The mutations of *pkd1* (85%) and *pkd2* (15%), which encode polycystin1 (PC1) and polycystin2 (PC2) respectively, cause the formation and enlargement of renal cysts in ADPKD [72]. Fifty percent of patients until the sixth decade of life reach ESRD with extrarenal manifestations, such as liver cysts, cerebral aneurysms, and cardiovascular disorders [73]. ADPKD is characterized by bilateral multiple fluid-filled cysts, damaged functional renal parenchymal, and reduced renal function, and is manifested with excessive proliferation of renal epithelium, secretion of cystic fluid and interstitial fibrosis [74]. Yet, only type-2 AVP receptor antagonist tolvaptan has been approved by FDA for treating ADPKD [75]. 

The pathophysiological mechanisms of ADPKD involves the abnormal intracellular calcium concentration and increased cAMP level, which activate repair signaling, including MAPK/ERK, PI3K/Akt, mTOR, Wnt/β-catenin, and so forth [76,77]. Although there is still a lack of satisfactory therapeutic drugs and the underlying mechanisms are not fully understood, AQPs have shown pivotal roles in the progression of ADPKD.

The first evidence of AQPs playing a role in ADPKD came from human samples [78]. Our group has reported that AQP1 mediated the inhibition of renal cyst development by restraining Wnt/β-catenin signaling in an orthologous ADPKD mice model [79] (Figure 5). The data presented that overexpressing AQP1 inhibited cyst growth in MDCK cyst model (Figure 5A). Contrarily, deletion of *AQP1* promoted cyst development in embryonic kidney and PKD mice (Figure 5B,C). In immunoprecipitation analysis, AQP1 was shown to interact with β-catenin, GSK3β, LRP6, and Axin1 (Figure 5D). Thus, the presence of AQP1 might directly cause the stability of destruction complex increased and promote the phosphorylation and degradation of β-catenin (Figure 5D), which suggests loss of AQP1 increases β-catenin accumulation and translocation into the nucleus to promote the transcription of Wnt target genes. AQP1 has great possibility to inhibit Wnt signaling and shows potential to retard the progression of renal cysts in clinic. 

Overexpression and mislocalization of AQP2 in the cytoplasm of collecting cells were observed in HIF-1α mutant mice, with renal cyst formation and urinary concentration defect [80]. In one study, steviol slowed cyst growth by reducing AQP2 expression and promoting AQP2 degradation in vitro [81]. This evidence shows potential therapeutic exploitation avenues for AQPs in ADPKD.

In addition, deletion of *Aqp11* caused proximal tubule cyst formation in the kidney, which results from the defective trafficking of PC1 [82]. AQP11-null mice have been an ideal model for studying proximal tubular cyst formation in ADPKD [83]. Therefore, the mutation of *AQP11* established a novel tool for studying renal cyst diseases.

## 7. AQP1 in Renal Cell Carcinoma

With the development of imaging technologies, the incidence of renal carcinoma increased in recent years. Renal cell carcinoma (RCC) is the most common neoplasm in the kidney and accounts for almost 3%~4% of adult malignant tumors [84]. About 30% of patients are unsuitable to be cured by surgical resection due to tumor metastasis [85]. Therefore, early diagnosis is pretty essential for treatment of RCC. A growing body of studies have identified that urine level of AQP1 is a sensitive, specific, reliable diagnostic biomarker for clear cell and papillary RCC [86,87,88]. Although the role of AQPs in the initiation and development of neoplasm has not been reported, cancer shares a lot in common with ADPKD, which sheds light on further research in renal cancer [89].

## 8. AQPs in Renal Fibrosis

Renal fibrosis is the common endpoint event of various CKDs and the extent of fibrosis indicates the residual renal function [90]. Fibrosis represents a series of dynamic processes involving imbalance between production and degradation of ECM, epithelial-to-mesenchymal transition (EMT), immune cell infiltration, and fibroblast activation [91]. Intervention in the progression of renal fibrosis highlights a promising therapeutic target for CKD. Loss of AQP1 was found in the unilateral ureteral obstruction (UUO) mice kidney, which could be reversed by EMT suppression, indicating AQP1 was involved in the process of EMT [92]. An in vitro study utilizing human proximal tubular epithelial cells showed the expression of AQP1 decreased in the model of aristolochic acid 1 (AA-1)-induced EMT and was regulated by ERK1/2 signaling, which implied that AQP1 might be a novel target for EMT [93]. 

In UUO animal models, it has been proven that the abundance of AQP2 was reduced partly leading to urinary concentrating defect [94]. Cholecalciferol cholesterol emulsion has been demonstrated to inhibit renal fibrosis through increasing AQP2 and decreasing AQP4 protein expression in UUO animal models [95]. Moreover, aliskiren restored the protein level of AQP2 by inhibiting inflammasome to exert its protective effect in the obstructed kidneys [94]. In future studies, inhibitors or drugs regulating the expression of AQPs may provide a new perspective for preventing fibrosis.

## 9. Other Diseases

In addition, analysis of three patients with autoimmune diseases showed autoantibodies targeting AQP2 or its upstream molecules caused tubulointerstitial nephritis [96]. Urinary increased excretion of AQP2 was mainly responsible for polyuria in patients with pyelonephritis and IgA nephropathy, suggesting AQP2 may be a treatment target for those diseases [97,98]. 

The reduced expression of AQP1, AQP2, and AQP4 was observed in mice with hydronephrosis [99]. Of note, although numerous studies have shown that the expression of AQPs was altered in renal diseases, the specific and direct role and function of AQPs, due to their permeability of water and other small molecules and regulation of intracellular signaling, require further study by the aid of AQPs knockout or overexpressing animal models.

## 10. Future Perspectives

An abundance of evidence has implied that AQPs are potential targets for a series of related disorders, such as NDI, ADPKD, edema, inflammatory disease and cancer [100]. However, the underlying mechanisms are not fully understood yet. In consideration of the permeability of AQPs, the effect of molecules crossing cell membranes depending on AQPs should be associated with the elucidation of their roles in renal diseases. Mice deficient in AQP1, AQP2, AQP3, and AQP4 exhibit diuresis. Restoration of these functional AQPs shows potential to treat NDI and inhibitors of these AQPs are predicted to be novel diuretics. The transportation of glycerol, H_2_O_2_ and other small molecules into cells might affect the intracellular signaling to regulate cellular proliferation, energy metabolism, and inflammation. Modulation of AQP-mediated small-molecule transport might be the drug target for AKI, PKD, renal fibrosis, and RCC. With clarification of AQP structure, computational methods have been used to screen AQP inhibitors for drug discovery. Because of the absence of gold-standard assays of AQP activity and the poor druggability of known small molecules, the development of AQP inhibitors progressed slowly [100]. So far, several compounds and pharmaceutical formulations have been proposed to modulate the function of AQPs and might provide new tools for studying related diseases [1]. Notwithstanding, the absence of effective AQP modulators hinders the development of therapeutic intervention and the translation from bench to bedside. In future studies, exploring highly selective, stable, and safe AQP inhibitors or activators will allow scientists to evaluate the physiological effects and therapeutic consequences. 

## Figures and Tables

**Figure 1 ijms-20-00366-f001:**
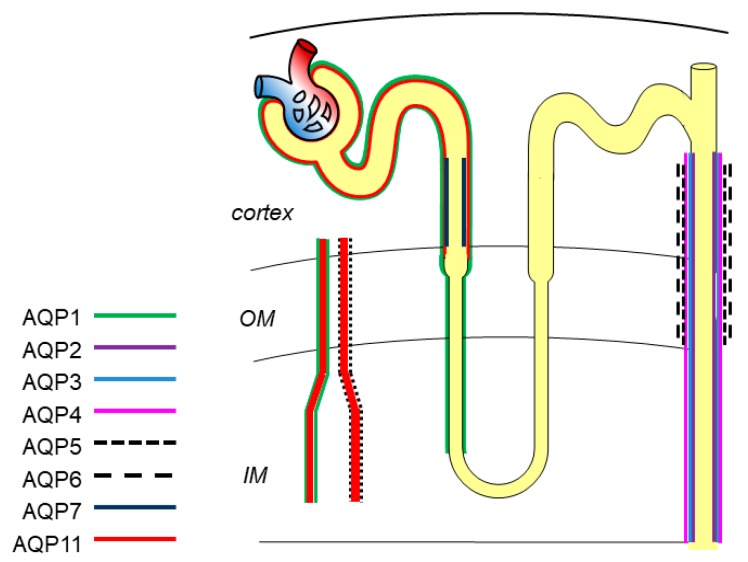
Expression localization of AQPs in kidney. AQP1 is located in proximal tubule, descending thin limbs of Henle, and vasa recta; AQP2, AQP3, AQP4, AQP5 and AQP6 are in the collecting duct; AQP7 and AQP11 are expressed in proximal tubule.

**Figure 2 ijms-20-00366-f002:**
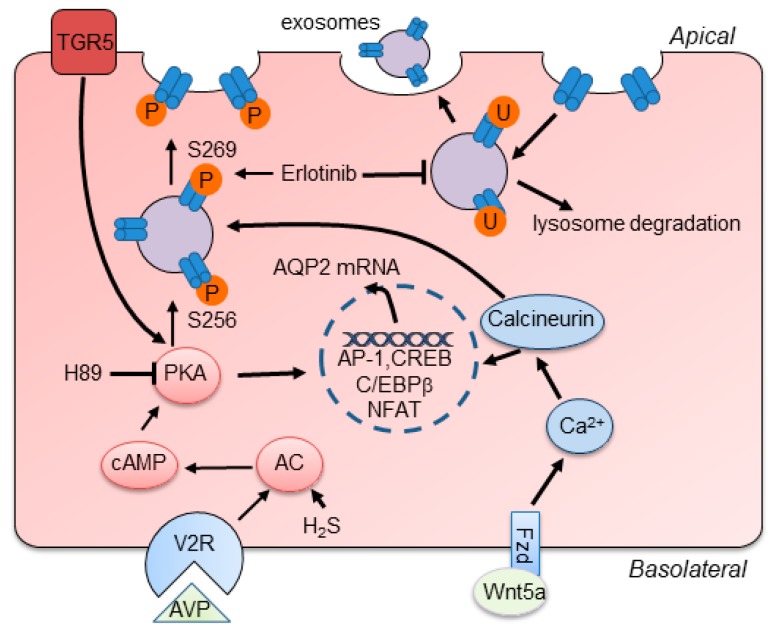
Schematic summary of main regulatory mechanisms of AQP2. AVP binding to V2R stimulates the activation of canonical cAMP/PKA signaling and increases the expression and phosphorylation of AQP2 at S256 and S269, leading to the apical membrane trafficking of AQP2. Activation of TGR5 increases the activation of PKA to induce the expression of AQP2. H_2_S increases AQP2 expression via enhancing the activation of cAMP/PKA signaling. Besides, Wnt5a binds to Fzd receptors and increases the level of intracellular calcium, which stimulates calcineurin and increases the expression and phosphorylation of AQP2. Erlotinib promotes AQP2 expression in the apical membrane by increasing the phosphorylation of AQP2 and reducing its endocytosis and degradation. The translocation of AP-1, CREB, C/EBPβ, and NFAT into the nucleus regulates the expression of AQP2.

**Figure 3 ijms-20-00366-f003:**
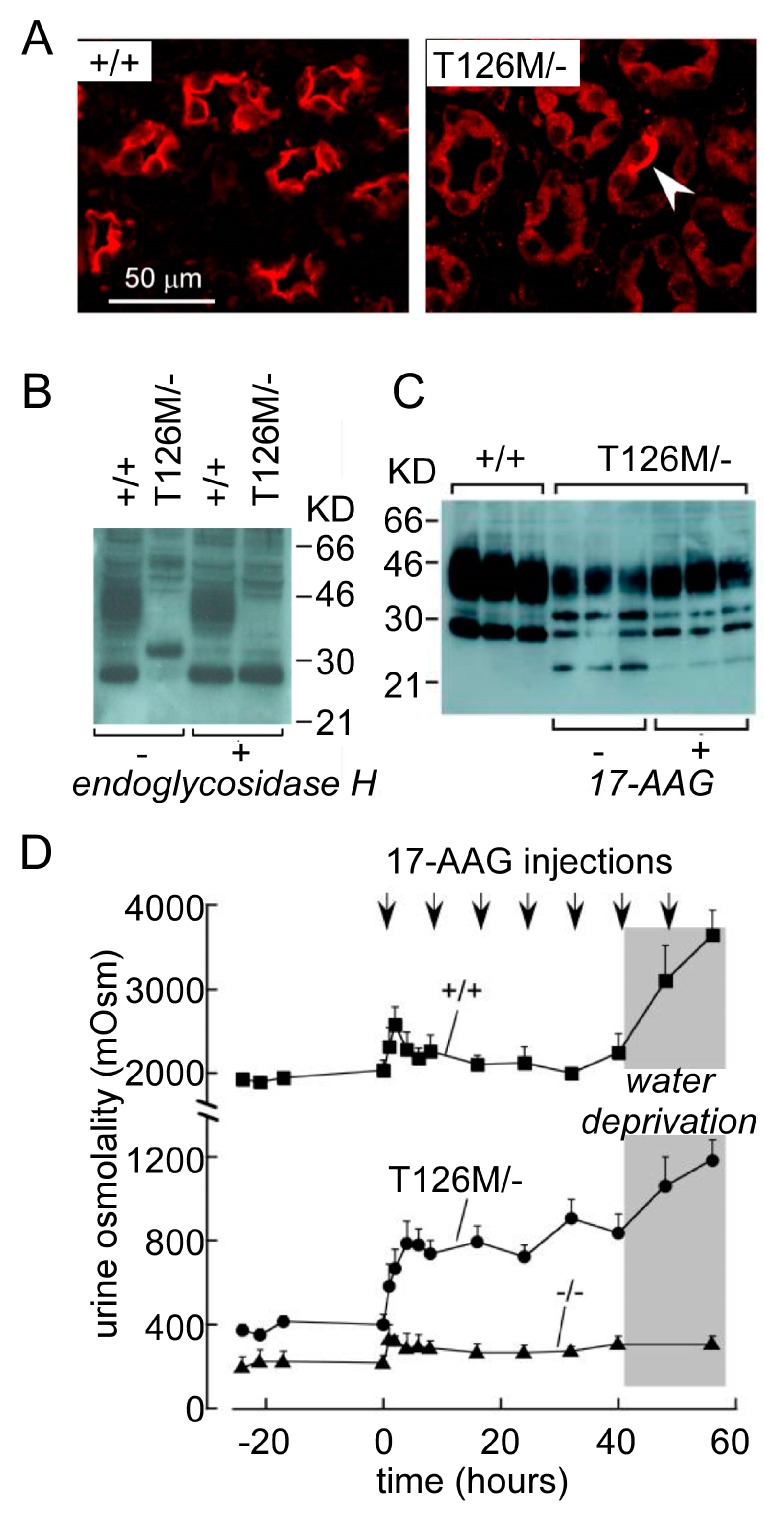
NDI caused by AQP2 mutation. (**A**) The immunofluorescence of AQP2 in kidneys of wild-type (left) and inducible AQP2^T126M/−^ mice (right). (**B**) Immunoblot of AQP2 from wild-type and inducible AQP2^T126M/−^ mice. (**C**) AQP2 protein immunoblot of kidney homogenates from wild-type and AQP2^T126M/−^ mice treated with or without Hsp90 inhibitor 17-AAG. (**D**) Urine osmolality in wild-type, AQP2 knockout, and inducible AQP2^T126M/−^ mice given free access to food and water before and after 17-AAG treatment. The block, solid ball, and triangle represent wild-type mice, AQP2^T126M/−^ mice, and AQP2^−/−^ mice, respectively (adapted from Ref [38]).

**Figure 4 ijms-20-00366-f004:**
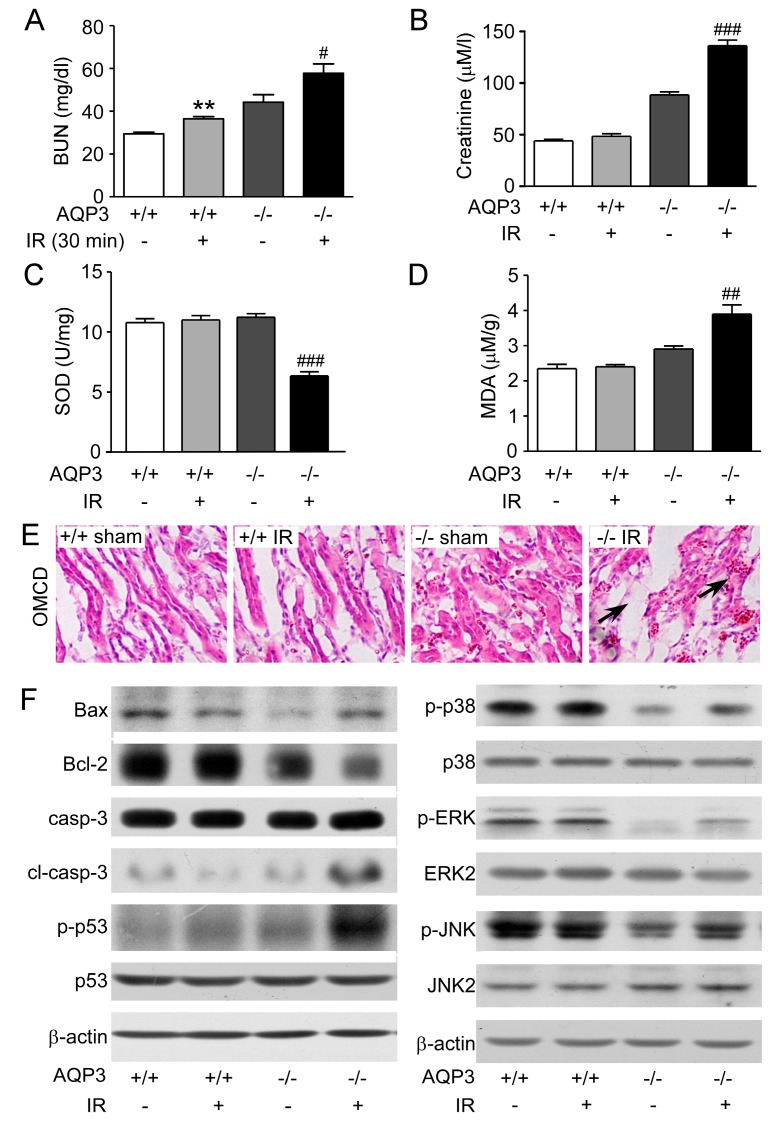
AQP3 deficiency aggravates IR injury. (**A**) Serum BUN level in wild-type and AQP3^−/−^ mice after reperfusion following 30 min ischemia. (**B**) Serum creatinine level in wild-type and AQP3^−/−^ mice after reperfusion following 30 min ischemia. (**C**) The activity of SOD in renal tissue. (**D**) The level of MDA in renal tissue. (**E**) Representative outer medullary collecting duct (OMCD) images of H&E staining of wild-type and AQP3^−/−^ kidneys after sham surgery or I/R. Black arrow represents dilated collecting ducts. (**F**) Different protein expression level of kidneys after reperfusion was detected by Western blot (adapted from Ref [56]).

**Figure 5 ijms-20-00366-f005:**
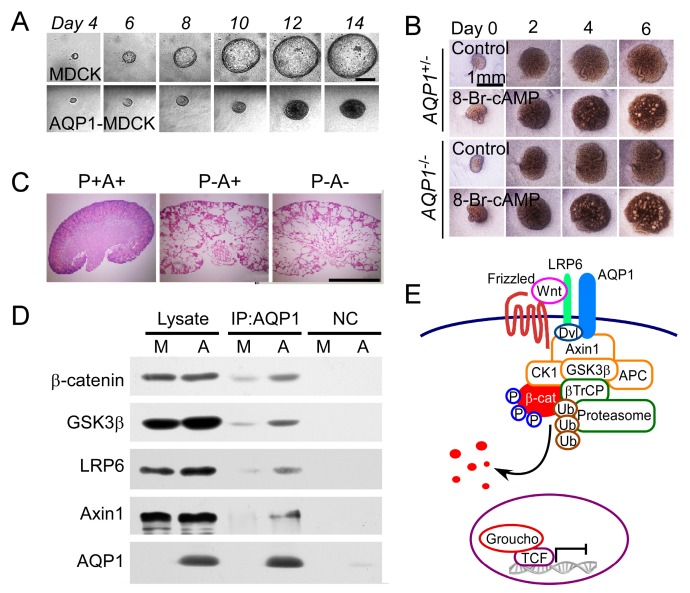
AQP1 retards renal cyst development. (**A**) Representative images of MDCK and AQP1-MDCK cyst from day 4 to day 14. (**B**) Representative images of AQP1^+/−^ and AQP1^−/−^ embryonic kidney cyst from day 0 to day 6. (**C**) Representative images of wild-type, PKD, and AQP1^−/−^ PKD kidneys. (**D**) Coimmunoprecipitation with anti-AQP1 showing the protein AQP1 interaction with β-catenin, GSK3β, LRP6, and Axin1 in AQP1-MDCK cells. M and A indicate MDCK cells and AQP1-MDCK cells, respectively. (**E**) Schematic of AQP1 regulating β-catenin (adapted from Ref [79]).

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
