# Peer review of "Aquaporins in Renal Diseases"

_ijms, 2019, doi:10.3390/ijms20020366_

Reviewer 1 Report

This is a well-balanced review describing the role of aquaporins in renal diseases. For all figure which are composites of data from other publications (Figs 3, 4, 5), the authors need to reference those publications in the figure legends, indicating which panel comes from which publication, and that they have permission to reproduce these figures.

Author Response

Point 1: This is a well-balanced review describing the role of aquaporins in renal diseases. For all figure which are composites of data from other publications (Figs 3, 4, 5), the authors need to reference those publications in the figure legends, indicating which panel comes from which publication, and that they have permission to reproduce these figures.

Response 1: Thanks for comments. We have cited the related reference of Fig 3, 4, 5 in the figure legends. And we have acquired copyright permission to reproduce these figures.

Reviewer 2 Report

The present manuscript examined the renal aquaporins (AQP) in normal and pathological conditions.

A huge amount of literature was examined and described in a clear and synthetic way. The review is well written, attractively presented and updated.

The Authors need to address the following point:

- Throughout the manuscript some abbreviations were present but not abbreviated when first appeared in the text and sometimes not present in the abbreviation list.

e.g, EGFR, BUN; ESRD abbreviated  NOT when first appeared.

- Some typos should be amended: e.g. pg. 8 line 262 “promote motes”; pg. 3 line 94 “PH” etc.

- pg. 1 lines 28-30; the sentence should be revised regarding the hydrogen peroxide permeability. AQP8 is an orthodox AQP but have high H2O2 permeability. Conversely, AQP7 and AQP10 have not been shown to have permeability to H2O2.

- Figure 1; AQP5 is not clearly shown in the figure: the dotted line is similar to that of AQP6.

- Pg. 2 line 57; please explain what Erlotinib is.

- Figure 3; symbols in Fig. 3 D were not present nor explained in the legend.

- Figure 4; arrows in panel E should be explained.

- Pg. 8 line 231; Why AQP5 is obviously increased?

- Pg. 8 line 235; change “David P et al “with “Choma et al”.

- Figure 5 legend; please describe M  and A.

- Perspectives should be better detailed or eliminated.

Author Response

The present manuscript examined the renal aquaporins (AQP) in normal and pathological conditions. A huge amount of literature was examined and described in a clear and synthetic way. The review is well written, attractively presented and updated. 

The Authors need to address the following point:

Point 1: Throughout the manuscript some abbreviations were present but not abbreviated when first appeared in the text and sometimes not present in the abbreviation list. 

e.g, EGFR, BUN; ESRD abbreviated NOT when first appeared.

Response 1: We have amended the abbreviations and present them in abbreviation list. All EGFR, BUN, ESRD, PKA and H2S have been showed full name when they first appeared.

 Point 2. Comment: Some typos should be amended: e.g. pg. 8 line 262 “promote motes”; pg. 3 line 94 “PH” etc.

Response 2:  The typos have been amended in our revised manuscript with highlight. Such as, “promote motes” has been corrected as “promote the”. “PH” has been corrected as “pH value”.

 Point 3: pg. 1 lines 28-30; the sentence should be revised regarding the hydrogen peroxide permeability. AQP8 is an orthodox AQP but have high H2O2 permeability. Conversely, AQP7 and AQP10 have not been shown to have permeability to H2O2.

Response 3: Basing on our knowledge, both AQP6 and AQP8 are orthodox AQPs, even their water permeability was not high. There is no solid evidence that AQP6 and AQP8 can transport other small molecules. Due to different aquaglyceroporins are selectively permeable to different solutes, we rewrite the sentence “Aquaglyceroporins, including AQP3, AQP7, AQP9, AQP10, are permeable to water and some small uncharged solutes, such as glycerol, and urea”. Meanwhile, we revised our reference [1].

 Point 4: Figure 1; AQP5 is not clearly shown in the figure: the dotted line is similar to that of AQP6.

Response 4: Figure 1 has been modified following the suggestion.

 Point 5: Pg. 2 line 57; please explain what Erlotinib is.

Response 5: Erlotinib, an EGFR inhibitor, has been explained in text.

 Point 6: Figure 3; symbols in Fig. 3 D were not present nor explained in the legend.

Response 6: The symbols have been explained in the legend of Figure 3.

 Point 7: Figure 4; arrows in panel E should be explained.

Response 7: “Black arrows represent dilated collecting ducts” has added in the legend of Figure 4.

 Point 8: Pg. 8 line 231; Why AQP5 is obviously increased?

Response 8: The data was cited from Reference [57], which showed that AQP5 was pathologically upregulated in the DN patients’ kidney samples.

 Point 9: Pg. 8 line 235; change “David P et al “with “Choma et al”.

Response 9: Changed.

Point 10: Figure 5 legend; please describe M and A.

Response 10: “M as MDCK cell and A as AQP1-MDCK cell” have been described in the legend of Figure 5.

 Point 11. Comment: Perspectives should be better detailed or eliminated.

Response 11: We replenish some details about the therapeutic targets of AQPs in Future Perspectives part.